# Efficacy and Safety of Sugammadex for the Reversal of Rocuronium-Induced Neuromuscular Blockade in Patients with End-Stage Renal Disease: A Systematic Review and Meta-Analysis

**DOI:** 10.3390/medicina57111259

**Published:** 2021-11-17

**Authors:** Young-Sung Kim, Byung-Gun Lim, Young-Ju Won, Seok-Kyeong Oh, Jung-Suk Oh, Soo-Ah Cho

**Affiliations:** Department of Anaesthesiology and Pain Medicine, Korea University Guro Hospital, Korea University College of Medicine, Seoul 08308, Korea; stelios@naver.com (Y.-S.K.); moma2@naver.com (Y.-J.W.); nanprayboy@naver.com (S.-K.O.); elf6728@naver.com (J.-S.O.); 93tndk@gmail.com (S.-A.C.)

**Keywords:** kidney failure, chronic, neuromuscular blockade, renal insufficiency, chronic, rocuronium, sugammadex

## Abstract

*Background and Objectives:* Sugammadex is widely used in anesthesia to reverse rocuronium-induced neuromuscular blockade (NMB). In patients with compromised kidney function, most drugs show alteration of their pharmacokinetic profile with reduced clearance. The purpose of this article is to examine the efficacy, pharmacokinetics, and safety of sugammadex in end-stage renal disease (ESRD) patients receiving general anesthesia, using a systematic review. *Materials and Methods:* The databases of PubMed, EMBASE, the Cochrane Library, Web of Science, Scopus, KoreaMed, and ClinicalTrials.gov were searched for studies comparing the efficacy or safety outcomes of sugammadex administration for the reversal of rocuronium-induced NMB, in ESRD patients (group R) or in those with normal renal function (group N) undergoing surgery under general anesthesia. *Results:* We identified nine studies with 655 patients—six prospective, case-control studies with 179 patients (89 and 90 in groups R and N) and three retrospective observational studies with 476 ESRD patients. In the six prospective studies, the times taken to reach a train-of-four ratio ≥0.9, 0.8, and 0.7 were significantly longer in group R than in group N (weighted mean difference [95% confidence interval] [min]: 1.14 [0.29 to 2.00], 0.9 [0.24 to 1.57], 0.89 [0.20 to 1.57], respectively). The total plasma clearance of sugammadex was significantly lower in group R than in group N. There was no significant difference in the incidence of NMB recurrence and prolonged time to recovery between the groups. In the three retrospective studies, the possibility of sugammadex-related adverse events appears to be insignificant. *Conclusions:* Sugammadex may effectively and safely reverse rocuronium-induced NMB in patients with ESRD, although the recovery to a TOF ratio of 0.9 may be prolonged compared to patients with normal renal function. Further studies are needed, considering the small number of studies included and the high heterogeneity of some of the results.

## 1. Introduction

Sugammadex (Bridion^®^, Merk Sharp and Dohme Corp., Kenilworth, NJ, USA) is a modified cyclodextrin designed to selectively encapsulate aminosteroidal neuromuscular blocking agents (NMBAs) such as rocuronium and vecuronium, which leads to the rapid reversal of neuromuscular block (NMB) [1,2].

Sugammadex administered to the blood rapidly encapsulates the NMBA, leading to an increased gradient in the concentration of NMBA between the neuromuscular junction and plasma; subsequently, the NMBA present at the neuromuscular junction is rapidly released into the blood, and rapid NMB reversal is achieved [3]. The sugammadex-NMBA complex produced is inactive and hydrophilic and is mainly excreted by the kidney. In addition, NMBAs such as rocuronium or vecuronium are excreted mainly through the kidneys [2,4]. In patients with severe renal impairment, the pharmacokinetics of both rocuronium and sugammadex are altered, and, thus, the NMB reversal by sugammadex can be unpredictable or incomplete [2]. Therefore, the U.S. Food and Drug Administration does not recommend sugammadex for patients with a creatinine clearance of <30 mL min^−1^ [5].

Nevertheless, the use of sugammadex is often observed in clinical practice for surgical patients with chronic kidney disease in various clinical situations, and some prospective case-control or retrospective studies and case reports regarding the use of sugammadex in patients with end-stage renal disease (ESRD) have been published [6,7,8]. To date, no systematic review regarding the use of sugammadex in patients with severe renal impairment has been reported, while there have been several meta-analyses showing the effectiveness, safety, and superiority of sugammadex, compared to cholinesterase inhibitors for NMB reversal in adult patients without organ dysfunction; a systematic review would need to take into account the results of the studies that reported the use of sugammadex in patients with ESRD and analyze their pooled data.

Herein, we aim to examine the efficacy, pharmacokinetics, and safety outcomes of sugammadex reversal for rocuronium-induced NMB in patients with ESRD (group R) undergoing surgeries under general anesthesia, compared to those with normal renal function (group N), using a systematic review framework. We hypothesize that the recovery after sugammadex administration might be more delayed in group R than in group N, but the incidence of adverse events might not differ between the two groups.

## 2. Materials and Methods

The systematic review and meta-analysis were performed according to the Cochrane Review Methods and the Preferred Reporting Items for Systematic Reviews and Meta-Analyses (PRISMA) recommendations [9]. The protocol for the study was registered with PROSPERO (registration number: CRD42020135822). The systematic review and meta-analysis were performed using the existing literature and data and did not involve new human data. Thus, the study was exempt from institutional review board approval.

### 2.1. Database and Literature Sources

We searched PubMed, EMBASE, the Cochrane Controlled Trials Register, and Cochrane Database on Systematic Reviews, Web of Science, Scopus, and KoreaMed databases from 1 January 1980 to 30 September 2020, using medical subject headings (MeSH) and free-text terms to identify articles that reported the efficacy, pharmacokinetics, and safety outcomes after administration of sugammadex for the reversal of rocuronium-induced NMB in patients with ESRD or severe renal impairment undergoing surgeries under general anesthesia. The following search terms were used for the search of each database: “renal insufficiency, chronic kidney disease, chronic renal disease, acute kidney injury, end-stage kidney disease, end-stage renal disease, neuromuscular blockade, sugammadex, Org 25969, gamma-cyclodextrins, and rocuronium”. Language restrictions were not imposed in our search (Appendix A).

After the initial electronic search, we evaluated the bibliographies from all identified studies and performed a manual search using Google Scholar. To identify unpublished or ongoing studies, we searched the World Health Organization International Clinical Trials Registry Platform and the ClinicalTrials.gov database. The identified articles were assessed individually for inclusion in the analysis.

### 2.2. Study Selection

Decisions regarding the inclusion of the studies in the analysis were made by two independent reviewers (BG Lim and YJ Won) and were based on predefined inclusion criteria. Studies were selected after being subjected to two levels of screening. At the first level, we screened the titles and abstracts of the identified studies. At the second level, we screened the full texts. Discrepancies between reviewers were resolved through discussion. The following studies were included in our systematic review: (1) prospective case-control studies in which (i) sugammadex was administrated for reversing moderate or deep NMB induced by rocuronium in patients with ESRD (group R) versus those with normal renal function (group N) undergoing surgeries under general anesthesia; (ii) the time taken for a train-of-four (TOF) ratio to recover to 0.9, 0.8, or 0.7 was evaluated; (iii) pharmacokinetic parameters, including total plasma clearance and plasma concentration of sugammadex or rocuronium, were evaluated; (iv) the prevalence of post-anesthetic adverse events, including hemodynamic instability, recurrence of NMB, desaturation, other clinical signs of the inadequate recovery of neuromuscular function, and drug (sugammadex or rocuronium) related adverse events, were evaluated; (2) retrospective studies in which sugammadex was administrated in order to reverse moderate or deep NMB induced by rocuronium in patients with ESRD.

### 2.3. Data Extraction

Two reviewers (BG Lim and YJ Won) independently extracted data from each study using a pre-specified data extraction form, a Microsoft Excel spreadsheet (Office 2016 professional edition; Microsoft Corp.). Any unresolved disagreements were reviewed and resolved in consultation with a third reviewer (SK Oh). The data extracted were as follows: (1) the demographic, clinical, and treatment characteristics of the included studies, including the name of the first author, year of publication, journal name, study design, and the number and characteristics of participants; (2) the mean and standard deviation of the times taken to reach a TOF ratio ≥0.9, 0.8, and 0.7; (3) the total plasma clearance and plasma concentration of sugammadex or rocuronium, and dichotomous data on the incidence of post-anesthetic adverse events including hemodynamic instability, recurrence of NMB, desaturation, other clinical signs of inadequate recovery of neuromuscular function (e.g., incidence of prolonged time to recovery of a TOF ratio of 0.9), and any drug (sugammadex or rocuronium) related adverse events.

### 2.4. Assessment of Methodological Quality

The methodological quality of the included studies was evaluated by two blinded reviewers (SK Oh and JS Oh). For quality assessment of the included trials, Risk Of Bias In Non-randomized Studies (ROBINS-I) was applied [10]. We evaluated the possible existence and direction of bias and whether it was likely to have an impact on the effects of interventions.

### 2.5. Statistical Analysis

The primary outcomes of this review were the times taken to reach a TOF ratio ≥0.9, 0.8, and 0.7, which were defined as the times from the start of administration of sugammadex to recovery of a TOF ratio ≥0.9, 0.8, and 0.7. The secondary outcomes were the total plasma clearance of sugammadex, the plasma concentration and total plasma clearance of rocuronium, and the incidence of post-anesthetic adverse events, including the incidence of significant changes in blood pressure and heart rate (e.g., hypotension, hypertension), recurrence of NMB, desaturation, and other clinical signs of the inadequate recovery of neuromuscular function (e.g., prolonged time to recovery of a TOF ratio of 0.9).

Continuous variables, including the times taken to reach a TOF ratio ≥0.9, 0.8, and 0.7, and total plasma clearance of sugammadex, plasma concentration, and total plasma clearance of rocuronium, were analyzed using the weighted mean difference (WMD) with a 95% confidence interval (CI). In the case of the time taken to reach a TOF ratio of 0.9, a WMD with a 95% CI >0 would indicate that the time was longer in group R than in group N. The incidence of post-anesthetic adverse events (recurrence of NMB or prolonged time to recovery of a TOF ratio of 0.9), which is a dichotomous variable, was analyzed using the pooled risk ratio with a 95% CI.

The random effects approach (inverse variance or Mantel–Haenszel) was chosen to allow for the expected heterogeneity among the included studies. This is because the patient’s knowledge of epidemiology and content suggested that the data collected from different study designs would not meet the assumptions of “fixed effects meta-analysis.” Statistical heterogeneity in each meta-analysis was assessed using Cochran’s Q test and I^2^ statistics. For the I^2^ statistics, the proportion of between-study inconsistency due to true differences between the studies, rather than differences due to random error or chance, was determined; values > 50% were considered to have significant heterogeneity. For Cochran’s Q test, a *p*-value of <0.1, was considered statistically significant. We used RevMan (version 5.3) and STATA (version 13.0) for these analyses.

A subgroup or sensitivity analysis was not performed because of the small number of included studies. The analysis of publication bias (used when there were at least 10 studies included) was not assessable for this meta-analysis, considering the small number of included studies.

## 3. Results

### 3.1. Identification of Studies

The database search yielded 965 articles (Figure 1). Of these, 952 publications were excluded because it was clear from the title and abstract that they did not fulfill the selection criteria. We obtained the full manuscripts for the remaining 13 articles and evaluated them. We identified potentially relevant studies and excluded four publications for the following reasons: (1) they were case reports (*n* = 3); (2) it aimed to investigate only dialyzability (*n* = 1). Hence, nine studies were included in our analysis (Figure 1).

### 3.2. Study Characteristics and Patient Demographics

The details of the selected studies are summarized in Table 1 and Table 2. We identified nine studies with 655 patients, including six prospective, case-control studies with 179 patients (90 patients with ESRD and 89 patients with normal renal function) and three retrospective, observational studies with 476 patients with ESRD who required preoperative renal replacement therapy.

Staals et al. published two articles, reporting on the same patients, in 2008 and 2010 [2,13]. Regarding post-anesthetic adverse events, de Souza et al. [11] defined the recurrence of NMB as a decrease in the TOF ratio below 0.9, after complete recovery was detected, and they monitored the arterial oxygen saturation (SaO_2_), blood pressure, and heart rate until 2 h after the administration of sugammadex. Panhuizen et al. [12] assessed data associated with patient safety, including post-anesthetic adverse events, heart rate, blood pressure, and laboratory data, as well as data associated with the physical examination of patients for four weeks after surgery. In the studies of Staals et al. [2,13], oxygen saturation was monitored for 7 h after the administration of sugammadex for group N and 24 h for group R; they assessed for clinical signs of recovery until 48 h after sugammadex administration and collected data about the vital signs, blood chemistry, and hematology analysis for 2–4 weeks after surgery. The recurrence of NMB was defined as a decrease in the TOF ratio to <0.9, after full recovery had been detected, or as a deterioration in the clinical signs of recovery from NMB. Min et al. [15] evaluated the safety and tolerability of sugammadex through a clinical assessment of adverse events and other safety measures, including vital signs, medical history, physical examination, 12-lead electrocardiography, and standard laboratory tests obtained at pre-specified time points throughout the study.

Pharmacokinetic parameters were evaluated in three trials, and each study used liquid chromatography–mass spectrometry to measure the plasma concentration of sugammadex and rocuronium, and the assays in the three trials were carried out in full compliance with Good Laboratory Practice regulations.

In one study (Min et al., 2017) [15], blood samples were obtained before sugammadex administration (pre-dose) through 48 h after sugammadex administration (post-dose) for group N, and pre-dose through day 10 (216 h) post-dose for group R (flexibility was included to extend pharmacokinetic assessment as needed for up to three additional samples [days 14, 18, and 21]). In another trial (Panhuizen et al., 2015) [12], plasma concentrations of rocuronium and sugammadex were assessed using blood samples pre-dose through 24 h post-dose for group N. For group R, blood samples were obtained pre-dose through 28 h post-dose. Unfortunately, the validity of the sugammadex bioanalytical data failed to reach quality standards since sample-to-sample carryover could not be ruled out and re-assay was not possible because of unavailable duplo samples and stability issues. Thus, in this study, all sugammadex bioanalytical data were considered invalid and could not be used for pharmacokinetic analysis.

In the last study (Staals et al., 2010) [2], for pharmacokinetic parameters, blood sampling was obtained pre-sugammadex, as well as 2, 3, 5, 10, 15, 20, 30, and 60 min and 2, 4, 6, 8, 12, 18, and 24 h after sugammadex administration. In group R, further blood samples were obtained at 36 and 48 h after sugammadex administration, and in the case of hemodialysis (within 72 h after the operation), additional blood samples were obtained pre- and post-dialysis.

### 3.3. Main Results: The Primary and Secondary Outcomes of the Included Prospective Studies

In total, three studies were analyzed for the time taken to reach a TOF ratio ≥0.9, 0.8 and 0.7, and all variables were significantly longer in group R than in group N, although the heterogeneity was high (WMD [95% CI] [min]: 1.14 [0.29 to 2.00]; I^2^ = 86%, 0.9 [0.24 to 1.57]; I^2^ = 87%, 0.89 [0.20 to 1.57]; I^2^ = 92%, respectively) (Figure 2).

The results of analysis on the pharmacokinetic parameters are as follows (Figure 3): Two studies [2,15] reported the total plasma clearance of sugammadex and rocuronium. The total plasma clearance of sugammadex was significantly lower in group R than in group N (WMD [95% CI] [mL min^−1^]: −87.18 [−136.34 to −38.01]; I^2^ = 0%). One study (Staals et al., 2010) [2] analyzed the total clearance of rocuronium. The total clearance of rocuronium was significantly lower in group R than in group N (MD [95% CI] [mL min^−1^]: −125.2 [−153.59 to −96.81]). Two studies [2,12] analyzed the plasma concentration of rocuronium after 12 h of sugammadex injection and found it was significantly higher in group R than in group N (WMD [95% CI] [ng mL^−1^]: 1023.32 [260.04 to 1786.6]; I^2^ = 97%). One study (Staals et al., 2010) [2] analyzed the plasma concentration of sugammadex after 6 h of sugammadex injection and found it was significantly higher in group R than in group N (MD [95% CI] [μg mL^−1^]: 7.7 [6.63 to 8.77]).

Regarding the safety outcomes, there was no significant difference in the incidence of recurrence of NMB or prolonged time to recovery of a TOF ratio to 0.9 between the two groups (risk difference [95% CI]: −0.01 [−0.07 to 0.04]; I^2^ = 0%, risk ratio [95% CI]: 2.87 [0.61 to 13.53]; I^2^ = 0%, respectively) (Figure 4).

For the other adverse events, three trials presented no clinically meaningful evidence (hemodynamic instability, such as a significant change in blood pressure, heart rate, and hypersensitivity) related to sugammadex administration [11,12,15]. Staals et al. [2] reported that two patients had low systolic pressure, and one patient had low diastolic pressure in group R, whereas in group N, one had high diastolic pressure, and one had low diastolic pressure. However, in all patients, the blood pressure changes were considered to be clinically insignificant and returned to baseline after anesthesia, and no markedly abnormal heart rates were observed. No laboratory abnormality related to sugammadex injection was reported in any of the studies, and there was no desaturation or other clinical signs of the inadequate recovery of neuromuscular function in any of the studies.

### 3.4. The Results of the Included Retrospective Studies

The results of post-anesthetic adverse events presented in the three retrospective observational studies with 476 patients with ESRD are as follows: Adams et al. [7] reported that there were 22 cases out of 158 patients (14%) with deferred tracheal extubation due to surgical or pre-existing medical conditions. Three of the 158 patients (2%) were re-intubated within 48 h postoperatively, but all of them were re-intubated due to their own medical problems and no incidence of recurrence of NMB after sugammadex injection was observed. This suggests that there is a very slim possibility of NMB recurrence after sugammadex injection in patients with ESRD. Paredes et al. [16] demonstrated that nine cases out of 219 patients (4.1%) were re-intubated, and of these, three (1.4%) patients were not excluded because of the possibility of sugammadex-related residual NMB. However, there was no mortality associated with sugammadex. Ono et al. [6] reported that there were no complications related to sugammadex administration in 99 patients.

### 3.5. Risk of Bias in the Prospective Case-Control Studies

Of the six prospective case-control studies, five studies were evaluated to have an overall “serious” risk of bias by ROBINS-I protocol, and only one study was evaluated as at an overall “moderate” risk of bias (Table 3).

## 4. Discussion

In ESRD patients, the clearance of rocuronium may decrease and the duration of action may be prolonged, and similar changes may occur in the pharmacokinetics of sugammadex. It is very important to predict changes in the effect, but there is no clear knowledge of this. In particular, the administration of sugammadex is not recommended in ESRD patients, and its safety has not been clearly proven until now [17], making it difficult to apply sugammadex in clinical practice and proceed with clinical studies. Therefore, the number of clinical studies reported to date is limited [2,11,12,13,14,15], and only a few case reports have been reported [8,18,19]. Nevertheless, the demand and need for the administration of rocuronium and sugammadex in ESRD patients in actual clinical practice is increasing, and several recently reported retrospective studies dispute this need [6,7,16,20]. Therefore, considering that there is an increasing need to evaluate the efficacy and safety of sugammadex in patients with ESRD, we tried, systematically and closely, to examine the efficacy and safety of the administration of rocuronium and sugammadex in ESRD patients, using a systematic review framework as far as this is possible.

This systematic review showed that the times taken to reach a TOF ratio ≥0.9, 0.8, and 0.7 were significantly longer in patients with ESRD, and the plasma clearance of sugammadex was significantly lower in patients with ESRD based on the meta-analysis results of the six prospective studies. However, given that the differences are not large enough to cause a clinically significant difference, the recovery of the patients could be slightly slower than that of patients with normal renal function. Moreover, there was no significant difference in the incidence of recurrence of NMB, in the delayed recovery to a TOF ratio of 0.9, or in other clinical signs of the inadequate recovery of neuromuscular function between patients with ESRD and those with normal renal function. In addition, the possibility of sugammadex-related residual NMB appeared to be insignificant in the three retrospective studies. Taken together, we suggest that sugammadex effectively and safely reverses rocuronium-induced NMB in patients with severe renal impairment, although further studies are needed to consider the small number of included studies and the high heterogeneity of some results.

In the results of the pharmacokinetic assessment, the effect of renal impairment on the total plasma clearance appeared to be smaller for rocuronium than for sugammadex. Staals et al. [2] suggested that in patients with ESRD, extrarenal clearance of rocuronium can occur, and there may still be a low concentration of rocuronium unbound and available for hepatic metabolism and elimination, even after the encapsulation of rocuronium by sugammadex. The greater effect of renal impairment on the total plasma clearance of sugammadex compared to that of rocuronium means that the plasma concentration of sugammadex remains relatively high in the postoperative period, suggesting that the potential risk of recurarization by unbound rocuronium may be low.

Although the plasma concentration of rocuronium after 12 h of sugammadex injection was significantly higher in patients with ESRD, this result is presumed to be due to the limitation of the assay method. In other words, an increase in the plasma concentration of rocuronium after the administration of sugammadex was detected because the assay method cannot differentiate between encapsulated and free rocuronium. Thus, it is not possible to determine the plasma concentration of unbound rocuronium [2,21,22]. Considering that there was no significant difference in the incidence of recurrence of NMB or prolonged time to recovery of a TOF ratio of 0.9 between patients with ESRD and those with normal renal function in this meta-analysis, and that there was no desaturation or other clinical signs of the inadequate recovery of neuromuscular function in any of the studies, the increased plasma concentration of rocuronium does not seem to be highly likely to be clinically harmful. However, there is a risk of potentially fatal complications such as recurarization, depending on how long the sugammadex–rocuronium complex exists in the blood and whether there is a change in its binding capacity. Thus, given that the sugammadex–rocuronium complex is retained in the body for longer in patients with ESRD [12] and no clinical data on its long-term disposition are yet available, further studies with a longer follow-up period should be conducted to determine whether the rocuronium–sugammadex complex can be safely eliminated or has an impact on safety in patients with ESRD.

Regarding the removal of the sugammadex–rocuronium complex in patients with ESRD, Cammu et al. [23] evaluated the dialyzability of sugammadex and the sugammadex–rocuronium complex in intensive care patients with severe renal impairment. They reported that, when calculating the reduction ratio (the extent of the plasma concentration reduction at the end of a dialysis episode when compared with before dialysis), mean reductions in the plasma concentrations of sugammadex and rocuronium were 69% and 75%, respectively, during the first dialysis episode, and approximately 50% during sequential dialysis episodes. On average, the dialysis clearance of sugammadex and rocuronium in blood was 78 and 89 mL min^−1^, respectively. They concluded that hemodialysis using a high-flux dialysis method is effective in removing sugammadex and the sugammadex–rocuronium complex in patients with ESRD. Therefore, in the case of ESRD patients who have undergone preoperative renal replacement therapy such as hemodialysis, if they undergo hemodialysis within 24–48 h after surgery, the sugammadex–rocuronium complex is effectively removed, thereby alleviating concerns about the occurrence of complications such as recurarization.

Still, the safety issues of sugammadex for ESRD patients without high-flux hemodialysis, who choose supportive care without dialysis or with low-flux hemodialysis, remain unclear. While the high-flux dialyzer has larger pores and allows diffusion of greater amounts of toxins and middle molecules, the low flow dialyzer is less permeable to large-sized complexes, such as the sugammadex–rocuronium complex. Therefore, patients with ESRD undergoing low flux hemodialysis were expected to have less effective clearance of sugammadex and the sugammadex–rocuronium complex than patients undergoing high flux hemodialysis [2,23].

The incidence of post-anesthetic adverse events was generally small in the included studies. There was no significant difference between the two groups in the incidence of recurrence of NMB or prolonged time to recovery of TOF ratio to 0.9. In addition, there were no clinically meaningful hemodynamic instabilities such as hypotension, bradycardia, or hypersensitivity related to sugammadex administration. Moreover, no laboratory abnormalities related to sugammadex injection, desaturation, or other clinical signs of inadequate recovery of neuromuscular function have been reported in any of the studies. However, considering that safety assessments, including the reporting of items and observation periods related to the adverse events, varied for each study, further larger prospective studies are needed in this area. Four of the six prospective, case-control studies described monitoring items and observation periods related to the safety outcomes in detail; but one of the other two studies was abstract, and there was no mention of safety outcomes, and the other was a pharmacokinetic study that did not perform neuromuscular monitoring. Considering these points, it is difficult to completely exclude the possibility of residual NMB in the absence of quantitative NMB monitoring, particularly across the range of sugammadex doses employed [17]. Nonetheless, considering the fact that even three retrospective observational studies assessing 476 patients with ESRD reported few adverse events, the incidence of adverse events related to sugammadex reversal for rocuronium-induced NMB may not be much higher in ESRD patients than in normal patients.

This review may be limited by the high heterogeneity in some results, especially the time taken to reach a TOF ratio ≥0.9, 0.8, and 0.7, which could have been caused by the differences in the degree of NMB followed by the dose of sugammadex administered in each study. Another limitation is that we could not assess publication bias because the number of included studies was small.

Recently, a preclinical article showing the renal protective effect of sugammadex in the ischemia-reperfusion rat model was reported [24]. Future clinical trials are expected to evaluate the renal effects of sugammadex itself. Although it may be too early to apply the results of this meta-analysis to ESRD patients in real clinical practice, we tried to present the best conclusions based on the results of the relevant clinical studies reported so far. Therefore, it may have sufficient significance as reference material for the consideration of future research directions and for use in clinical settings.

In conclusion, sugammadex may effectively and safely reverse rocuronium-induced NMB in patients with ESRD, although the recovery time to a TOF ratio of 0.9 may be longer in patients with ESRD than those with normal renal function. However, further prospective or retrospective studies including more robust data, especially safety data are needed, considering the small number of studies included, the high heterogeneity of some results, and the insufficient safety information to support the recommended use of sugammadex in this population.

## Figures and Tables

**Figure 1 medicina-57-01259-f001:**
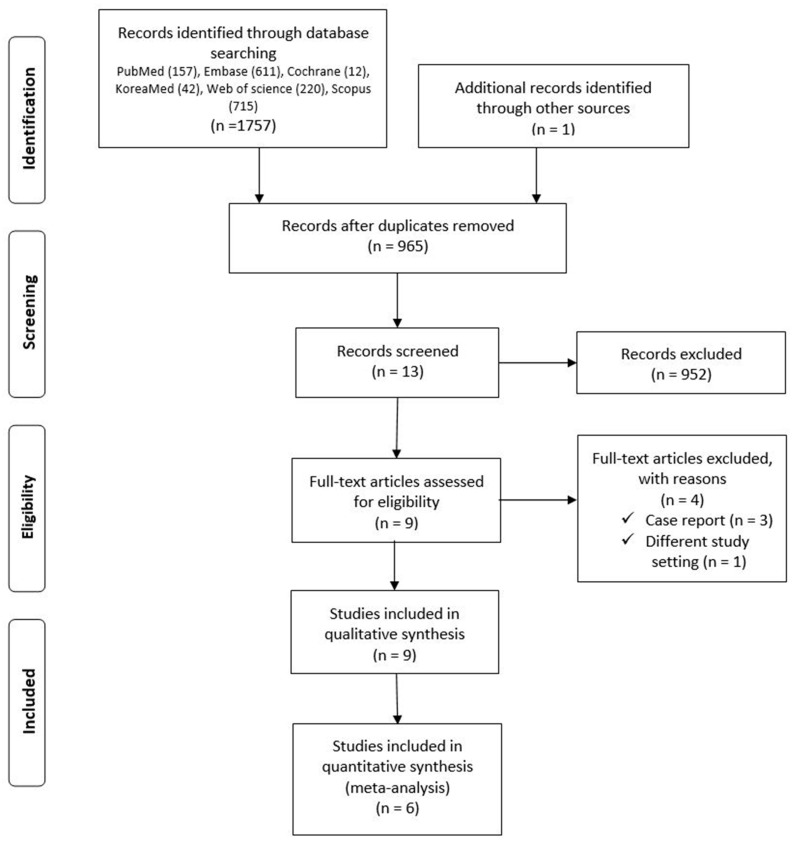
PRISMA flow diagram. The diagram shows the study selection process and provides reasons for exclusion of the records screened. PRISMA = Preferred Reporting Items for Systematic Reviews and Meta-Analyses.

**Figure 2 medicina-57-01259-f002:**
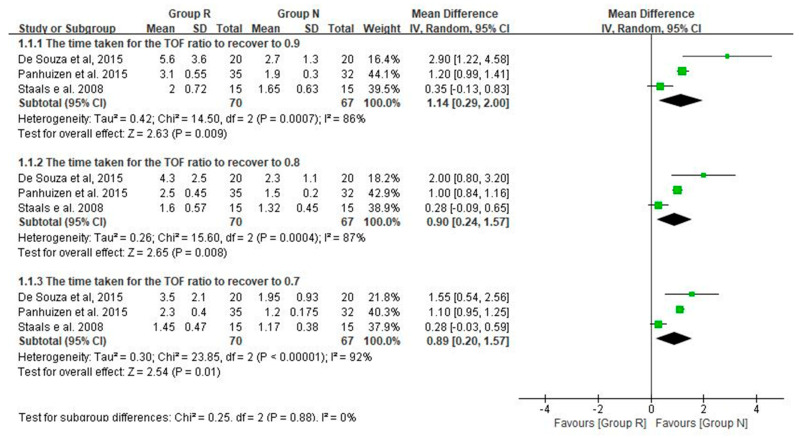
The time taken to reach a TOF ratio ≥ 0.9, 0.8, and 0.7 (min). SD = standard deviation; IV = inverse variance; CI = confidence interval.

**Figure 3 medicina-57-01259-f003:**
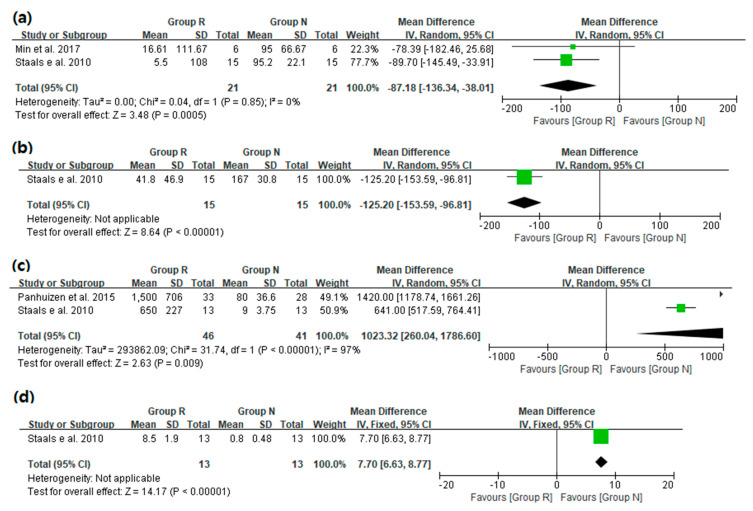
Pharmacokinetic parameters: (**a**) the total plasma clearance of sugammadex (mL min^−1^), (**b**) the total plasma clearance of rocuronium (mL min^−1^), (**c**) the plasma concentration of rocuronium 12 h after sugammadex injection (ng mL^−1^), (**d**) the plasma concentration of sugammadex 6 h after sugammadex injection (μg mL^−1^). SD = standard deviation; IV = inverse variance; CI = confidence interval.

**Figure 4 medicina-57-01259-f004:**
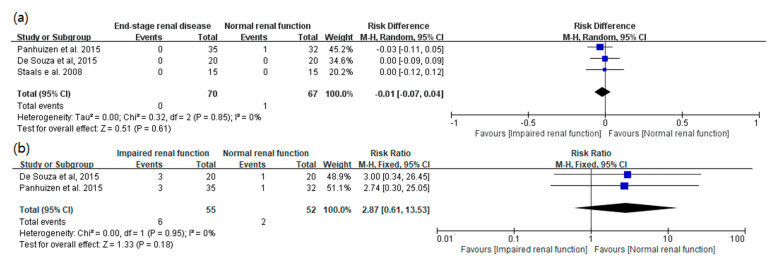
(**a**) Incidence of recurrence of neuromuscular blockade. (**b**) Incidence of prolonged time to recovery of a train-of-four ratio to 0.9. M-H = Mantel-Haenszel; CI = confidence interval.

**Table 1 medicina-57-01259-t001:** Characteristics of the included prospective case-control studies.

Study	Journal	Study Design	Center/Country	Group R (*n*)	Group *N* (*n*)	Age	Sugammadex Dose	NMB Monitoring
de Souza et al., 2015 [11]	*European Journal of Anaesthe-siology*	Prospective clinical trial	Two hospitals Brazil, and Spain	ClCr < 30 mL min^−1^ (20)	ClCr > 90 mL min^−1^ (20)	18–65	4 mg kg^−1^	Acceleromyography at the adductor pollicis muscle
Panhuizen et al., 2015 [12]	*British Journal of Anaesthesia*	Case control comparative study	Eight centers in Europe	ClCr < 30 mL min^−1^ (35)	ClCr ≥ 80 mL min^−1^ (35)	≥18	4 mg kg^−1^	Acceleromyography at the adductor pollicis muscle
Staals et al., 2008 [13]	*British Journal of Anaesthesia*	Prospective clinical trial	Three centers in Europe	ClCr < 30 mL min^−1^ (15)	ClCr ≥ 80 mL min^−1^ (15)	≥18	2 mg kg^−1^	Acceleromyography at the adductor pollicis muscle
Staals et al., 2010 [2]	*British Journal of Anaesthesia*	Prospective clinical trial	Three centers in Europe	ClCr < 30 mL min^−1^ (15)	ClCr ≥ 80 mL min^−1^ (15)	≥18	2 mg kg^−1^	Acceleromyography at the adductor pollicis muscle
Maeyama et al., 2014 [14]	*European Journal of Anaesthesiology*	Prospective clinical trial	University hospital, Japan	ClCr < 15mL min^−1^ (13)	ClCr > 90 mL min^−1^ (14)	≥18	4 mg kg^−1^	Not mentioned
Min et al., 2017 [15]	*International Journal of Clinical Pharmacology and Therapeutics*	Open label, two parts, phase 1 study	Clinical pharma-cology of Miami, USA	ClCr < 30 mL min^−1^ (6)	ClCr ≥ 80 mL min^−1^ (6)	≥18	4 mg kg^−1^	None

R: patients with end-stage renal disease; Group N: patients with normal renal function; NMB: neuromuscular blockade; ClCr: Creatinine clearance.

**Table 2 medicina-57-01259-t002:** Characteristics of the included retrospective observational studies.

Study ID	Journal	Study Design	Center/Country	Enrolled Criteria (*n*)	Age	NMB Reversal Agent
Adams et al., 2020 [7]	*Anaesthesia*	Two centers retrospective study	Pittsburgh Medical Center, Memorial Sloan Kettering Cancer Center, USA	End-stage renal disease which is mandatory for renal replacement therapy (158)	≥18	sugammadex
Paredes et al., 2020 [16]	*Canadian Journal of Anaesthesia*	Historical cohort study, three-distinct geographic locations	Scottsdale, AZ, Jacksonville, FL, Rochester, MN, USA	eGFR value < 15 mL min^−1^ (219)	≥18	sugammadex
Ono et al., 2018 [6]	*Journal of Anesthesia Clinical Reports*	Retrospective study	Aichi Medical University, Nagakute, Japan	Diagnosed severe with renal failure and underwent renal transplantation (99)	not mentioned	sugammadex

eGFR = estimated glomerular filtration ratio. NMB = neuromuscular blockade.

**Table 3 medicina-57-01259-t003:** Risk of bias assessment of prospective case-control studies for meta-analysis by ROBINS-I* protocol.

Study	Confounding	Selection	Classification of Interventions	Deviation from Interventions	Missing Data	Measurement of Outcomes	Selection of the Reported Result	ROBINS-I Overall
de Souza et al., 2015 [11]	Moderate	Moderate	Moderate	Moderate	Moderate	Serious	Low	Serious
Panhuizen et al., 2015 [12]	Moderate	Low	Moderate	Low	Moderate	Serious	Low	Serious
Staals et al., 2008 [13]	Moderate	Moderate	Low	Low	Low	Serious	Low	Serious
Staals et al., 2010 [2]	Moderate	Moderate	Low	Low	Moderate	Serious	Low	Serious
Maeyama et al., 2014 [14]	Moderate	Serious	Low	Low	Low	Low	Low	Serious
Min et al., 2017 [15]	Moderate	Low	Low	Low	Low	Moderate	Moderate	Moderate

## Data Availability

Not applicable.

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
