# Peer review of "Efficacy and Safety of Sugammadex for the Reversal of Rocuronium-Induced Neuromuscular Blockade in Patients with End-Stage Renal Disease: A Systematic Review and Meta-Analysis"

_medicina, 2021, doi:10.3390/medicina57111259_

Round 1
Reviewer 1 Report
Dear editors:
It is a great honor and pleasure for me to be invited as the reviewer for this important work. Kim et al. comprehensively reviewed the efficacy and safety of sugammadex for the reversal of rocuronium-induced neuromuscular blockade in patients with end-stage renal disease. This study topic is interesting and novel, attributing to Prof. Lim’s long-term efforts and contributions in this scientific field. Although the article is well-written that refreshes our understanding of Sugammadex, I have some comments concerning this study:
With end-stage renal disease (ESRD), patients need dialysis or a kidney transplant to stay alive. But patients can also choose to opt for conservative care to manage their symptoms. In light of current evidence-based medicine, the safety of sugammadex for the reversal of rocuronium-induced neuromuscular blockade in ESRD patients results from the dialysable property, particularly in high-flux hemodialysis. The safety issue for ESRD patients without high-flux hemodialysis remains unclear. Authors should highlight and address this critical issue.
Thank you for giving me the opportunity to review this interesting article. After minor revision, this important review article should be published as soon as possible.
Author Response
Thank you very much for your considerate review.
I fully agree with your opinion. The manuscript was revised as below.
Line 372-379
Still, the safety issues of sugammadex in ESRD patients without high-flux hemodi-alysis, who choose supportive care without dialysis or with low-flux hemodialysis, re-main unclear. While high-flux dialyzer has larger pores and allow diffusion of greater amounts of toxins and middle molecules, low flow dialyzer is less permeable to large-sized complexes, such as sugammadex-rocuronium complex. Therefore, patients with ESRD undergoing low flux hemodialysis were expected to have less effective clear-ance of sugammadex and the sugammadex-rocuronium complex than patients undergo-ing high flux hemodialysis.[23,24]
- Staals, L.M.; Snoeck, M.M.; Driessen, J.J.; van Hamersvelt, H.W.; Flockton, E.A.; van den Heuvel, M.W.; Hunter, J.M. Reduced clearance of rocuronium and sugammadex in patients with severe to end-stage renal failure: a pharmacokinetic study. Br J Anaesth 2010, 104, 31-39, https://dx.doi.org/10.1093/bja/aep340.
Thank you again your effort to review the paper.

Reviewer 2 Report
Kim et al submit a review paper entitled "Efficacy and Safety of Sugammadex for the Reversal of Rocuronium-Induced Neuromuscular Blockade in Patients with End-Stage Renal Disease: A Systematic Review and Meta-Analysis" In this interesting and timely review, they study the efficacy and secondary effects of Sugammadex for reversal of Rocuronium-Induced Neuromuscular Blockade in the context of End-Stage Renal Disease They conclude that Sugammadex may effectively and safely reverse rocuronium-induced NMB in ESRD, with a slighly delayed recovery.
The administration of sugammadex is not recommended in ESRD patients, although there is a demand. The review is thus uiseful. The design is very thorough, adn thje mat and meth well explained. The results are well written.
I think that some data if available on preclinical models would be useful in the introduction.
In the disucssion, a recapitulatve figure would be useful for the reader.
Author Response
Thank you for your kind review.
As your advice, I found an interesting preclinical study that recently published.
Line 404-406
Recently, a preclinical article showing the renal protective effect of sugammadex in the ischemia-reperfusion rat model was reported.[25] Future clinical trials are expected to evaluate the renal effects of sugammadex itself.
- Tercan, M.; Yilmaz Inal, F.; Seneldir, H.; Kocoglu, H. Nephroprotective Efficacy of Sugammadex in Ischemia-Reperfusion Injury: An Experimental Study in a Rat Model. Cureus 2021, 13, e15726, https://dx.doi.org/10.7759/cureus.15726.
Instead of additional figure, I added a supplementary table to summarize the results of this review.
Table S2. Summary of this review
|
ESRD patients’ changes in this review |
|
Delayed recovery to a TOF ratio of 0.9 Decreased clearances of both rocuronium and sugammadex Increased remaining plasma concentrations of both rocuronium and sugammadex No changes on the incidence of recurrence of NMB, inadequate recovery of neuromuscular function Longer retention time of sugammadex-rocuronium complex, but effectively removed through hemodialysis within 24-48 hours after surgery |
Thank you again for your effort to review our paper.
